# The Modified Phenanthridine PJ34 Unveils an Exclusive Cell-Death Mechanism in Human Cancer Cells

**DOI:** 10.3390/cancers12061628

**Published:** 2020-06-19

**Authors:** Malka Cohen-Armon

**Affiliations:** Sackler Faculty of Medicine, and Sagol School of Neuroscience, Tel-Aviv University, Tel-Aviv 69978, Israel; marmon@tauex.tau.ac.il

**Keywords:** the PARP inhibitor PJ34, distorted mitotic spindles, exclusive eradication of human cancer cells

## Abstract

This overview summarizes recent data disclosing the efficacy of the PARP inhibitor PJ34 in exclusive eradication of a variety of human cancer cells without impairing healthy proliferating cells. Its cytotoxic activity in cancer cells is attributed to the insertion of specific un-repairable anomalies in the structure of their mitotic spindle, leading to mitotic catastrophe cell death. This mechanism paves the way to a new concept of cancer therapy.

## 1. Background

In the last twenty years the modified phenanthridine PJ34 has been known for its activity as a PARP (polyADP-ribose polymerase) inhibitor [1,2] (Figure 1). Recently, PARP inhibitors attract the attention of researchers and clinicians due to their FDA approval for cancer therapy [3,4,5,6]. Several comprehensive overviews on the family of PARP proteins and their inhibitors have been published [3,7,8,9,10,11]. This overview is dedicated to the recently disclosed exceptional cytotoxicity of the PARP inhibitor PJ34 in human cancer cells, which does not affect human healthy cells.

The modified phenanthridine PJ34 is a stable molecule, fairly soluble in water (22 mg/mL), and permeable in the cell membrane. The potency of PJ34 to inhibit PARP proteins has been measured [8]. PJ34 is a potent inhibitor of PARP1 and PARP2 (approximate IC50, 20 nM). In addition, PJ34 inhibits tankyrase-1 and tankyrase-2 (approximate IC50, 1 μM). PJ34 hardly inhibits other PARP proteins [8]. PJ34 was originally invented to protect cells from cell death imposed by pathological stress conditions, such as ischemia [1,2,12,13,14].

The common concept in utilizing PARP inhibitors for cancer therapy is based on evidence associating PARP1 inhibition with the prevention of DNA break repair causing apoptotic cell death [6,15,16,17,18]. PARP1 is activated and polyADP-ribosylated in response to DNA single strand breaks frequently caused under stress conditions, including X-ray irradiation and application of DNA-damaging agents [3,4,5,6]. The binding of PARP1 to DNA single-strand breaks and its activation (polyADP-ribosylation) initiates their repair [3,4,5,6]. In addition, PARP1 is implicated in the alternative non-homologous end joining (A-NHEJ) mechanism of double strand DNA break repair [19,20]. Cancer cells have a higher percentage of DNA breaks relative to healthy cells, due to failure in arresting mitosis of cancer cells with damaged DNA [15,16]. In addition, DNA-damaging treatments are used for cancer therapy [6,16,17]. Thus, prevention of DNA repair by PARP inhibitors can be beneficial for cancer therapy, either as a monotherapy, or in combination with treatments causing DNA damage [11,16,17,18,19]. According to this concept, PARP inhibitors are offered to cancer patients carrying *BRCA* gene mutations (about 2% of the western world population). The BRCA protein is implicated in the repair of double-strand DNA breaks [17,21]. Mutations in BRCA impact the functioning of the BRCA protein, and increase DNA breaks in the cells of BRCA mutant carriers. This frequently increases the probability of mutations associated with malignancy [17,21]. Treatment with PARP inhibitors in BRCA mutant carriers is based on the interference of PARP1 inhibition with DNA repair in cells carrying a damaged DNA.

The partial efficacy of PARP inhibitors in the small population of *BRCA* mutant carriers, the resistance of a variety of cancer types to the currently approved PARP inhibitors, their side effects and the unclear impact of their chemical structure on their potency [21,22,23,24,25,26], urged a further investigation of the activity of PARP inhibitors in cancer therapy. It was observed that apart from PARP inhibition, some of these molecules target a variety of kinases implicated in signal transduction pathways in both healthy and malignant cells [27]. Unexpectedly, this research also disclosed that a group of phenanthrene derivatives acting as PARP inhibitors, exclusively kill human cancer cells without affecting benign cells [28,29,30,31,32]. Unlike other PARP inhibitors, these small molecules exclusively eradicated a variety of human cancer cells without affecting proliferating and non-proliferating healthy somatic cells. They did not affect human epithelial, mesenchymal and endothelial cells [28,29,30,31,32,33,34,35], nor healthy cells of mouse origin, including mouse embryonic fibroblasts (MEF), fibroblasts, neurons in the central nervous system and neuronal progenitor stem cells [28,29,31,32,33,34]. Their exclusive cytotoxic activity in human cancer cells was not shared by other potent PARP inhibitors [29,30,31]. Moreover, their toxicity in human cancer cells was independent of the expression of PARP1 and P53, PARP1 activity and DNA damage [29,34,35]. On the other hand, their exclusive cytotoxic activity in human cancer cells resembles the cytotoxic activity of other phenanthridines [36,37,38]. The modified phenanthridine PJ34, one of the molecules in this group, was the most potent in a variety of human cancer cells, including cells that are resistant to currently offered therapies [28,29,30,31,32]. Its specific cytotoxic activity in human cancer cells is summarized in this overview.

## 2. PJ34 Efficiently Eradicates a Variety of Human Cancer Cells in Tissue Cultures

After years of research based on PJ34-induced PARP inhibition in a variety of cell types under pathological conditions [1,2,8], additional activities of PJ34 have been disclosed. It was observed that PJ34 causes an irreversible cell growth arrest in cancer cells, that it interferes with angiogenesis, and, most interestingly, that PJ34 exclusively eradicates human cancer cells [29,30,39,40].

Incubation with PJ34 at higher concentrations than those inhibiting PARP1 (10–20 μM PJ34), completely eradicated within 48 h human MCF-7 breast cancer cells that are resistant to doxorubicin [28]. Furthermore, PJ34 (20–30 μM) eradicated within 72–96 h cancer cell types that are resistant to other therapies, including types of triple negative breast cancer, pancreatic cancer, ovary cancer, colon cancer and non-small lung cancer [28,29,30,31,32].

Gangopadhyay and colleagues found that incubation with 30 μM PJ34 for 72 h eradicates several human metastatic lung cancer cell lines: Calu-6, A549 and H460 [41]. In addition, PJ34, at higher concentrations than those inhibiting PARP1, arrested the growth of human liver cancer cell lines (HepG2 and SMMC7721) [42], and the human multiple myeloma RPMI8226 cell line [43]. PJ34 acts as a potent anti-proliferating agent in human leukemia cell lines (ATLL and transformed HTLV-I) [44], and in human ovarian cancer epithelial cells (C13 cell line) [45]. The cell death-inducing efficacy of PJ34 at higher concentrations than those inhibiting PARP activity has been also reported in a variety of breast cancer cell-lines, carrying or not BRCA mutations, and in a variety of triple-negative breast cancer cell-lines [46], as well as in melanoma cell lines and melanoma metastases [47], thyroid cancer cell lines [48], HeLa cells [49] and several glioblastoma cell lines [50]. PJ34 also efficiently prevented Helicobacter-induced gastric pre-neoplasia [51]. On the other hand, healthy proliferating cells treated with PJ34, at the same concentrations eradicating cancer cells, continued to proliferate in the presence of PJ34 as untreated cells for weeks [28,29,30,31,32]. Furthermore, incubation with PJ34 (20 μM) did not affect retinoic acid-induced differentiation in the human neuroblastoma cell line SHSY5Y [52], nor impaired the neuronal excitability of mouse hippocampal neurons [34].

Notably, mice treated with PJ34 for 2–3 weeks, continued to gain a similar amount of weight as untreated mice, and did not exert any visible stress or discomfort signs, as described below.

## 3. PJ34 Causes Eradication of Human Cancer Cells in Xenografts

The cytotoxicity of PJ34 in human cancer cells was tested in animal models as well. PJ34 (10 mg/kg) intraperitoneal (IP) injection 3 times a week for 3 weeks attenuated the growth of intracranial tumors of glioblastoma in nude mice [50]. The efficacy of daily treatment of PJ34 (30 mg/kg) using intraperitoneal injection over 14 days has been tested in xenografts of ovarian cancers. Treatment with PJ34 at this dosage most efficiently decreased the tumors’ size [53]. In contrast, treatment with PJ34 in xenografts prepared from uterus and ovarian cancer of BRCA carriers showed insufficiency when PJ34 was applied per os (10 mg/kg twice a day for 16 days), in comparison to chemotherapy with carboplatin (80 mg/kg) and paclitaxel (24 mg/kg) injected IV (intravenous) once a week [54].

PJ34 prevented the development of human breast cancer MCF-7 and triple negative breast cancer MDA-MB-231 tumors in immunocompromised mice treated with a slowly released PJ34 from subcutaneous osmotic pump (Alzet) over 14 days [28]. The PJ34-treated mice that were injected subcutaneously with MCF-7, 5 × 10^6^ cells before treatment, did not develop breast cancer tumors, and remained tumor-free during the following four months. In mice subcutaneously injected with human MDA-MB-231 triple negative breast cancer cells (5 × 10^6^), PJ34 slow release for 14 days prevented the development of tumors in three out of five mice, and those mice remained tumor-free during the following four months. Importantly, the treatments with PJ34 did not affect the vitality, growth and weight-gain of the treated mice during the follow-up periods [28]. The therapeutic potency of PJ34 was also tested in triple negative MDA-MB-231 breast cancer xenografts after tumor development. In these experiments PJ34 (60 mg/kg) was injected IP after the tumors reached a volume >100 mm^3^. PJ34 injected daily for 14 days efficiently suppressed tumor growth [31].

The efficacy of PJ34 (60 mg/kg, IV injected daily, 5 days a week, 14 intravenous injections) was tested in xenografts of human pancreas ductal adenocarcinoma PANC1 developed in immunocompromised mice. A substantial reduction of 80–90% in human cancer cells in the tumors was measured by immunohistochemistry in slices prepared from the excised tumors (16 mice) 30 days after the treatment with PJ34 has been terminated. One of the tumors disappeared after the treatment with PJ34 [32]. Benign fibroblasts infiltrated into the PANC1 tumors (stroma cells) were not impaired by the treatment with PJ34. Growth, weight-gain and behavior of the treated mice were not impaired during, and 30 days after the treatment with PJ34 has been terminated [32]. A similar cytotoxic activity of PJ34 was observed in patients-derived pancreatic cancer cells, and in patient-derived xenografts [32].

A combined treatment of PJ34 (IV) with other agents was examined, as well. A combined treatment of PJ34 with agents inducing TRAIL-mediated apoptosis in glioma xenografts reduced glioma tumor growth, and revealed minimal cytotoxicity in non-neoplastic astrocytes [55]. The combined treatment of PJ34 with TRAIL agonists was non-toxic to normal human primary glial and neuronal cells, anticipating minimal side-effects of this treatment in patients [55]. A similar effect was achieved in pancreas cancer xenografts treated with PJ34 in combination with agonists activating TRAIL-mediated apoptosis [56].

A combined treatment with PJ34 and the HDAC (histone-deacetylase) inhibitor SAHA blocked the growth of liver tumors [42]. In these experiments, HepG2 (5 × 10^6^) cancer cells were injected subcutaneous into immunocompromised mice. These mice were treated with a combination of PJ34 (IP, 10 mg/kg) and SAHA (IP, 25 mg/kg), 3 times a week for 3 weeks. Under these conditions, tumor growth was substantially suppressed without affecting human hepatocytes [42]. Notably, SAHA at higher doses was cytotoxic to normal liver human fetal hepatocytes, while high dosage of PJ34 did not harm these cells [42].

Combining PJ34 with other anti-cancer treatments enabled reducing their cytotoxic doses and achieved efficient eradication of HL-60, Jurkat-T cells, multiple myeloma cellsPRMI8226/R and B16F10 melanoma cells [57,58,59].

In view of these findings, the possibility that common mechanisms are targeted by PJ34 in a variety of human cancer cell types has been examined.

## 4. The Mechanism of Action of PJ34 in Human Cancer Cells

In a search for the mechanism of action of PJ34 in human cancer cells, several mechanisms have been considered. The activity of PJ34 in cancer cells has been suggested to promote cell death by preventing PARP1-mediated DNA repair. This suggested mechanism is in line with PARP inhibition sensitizing cancer cells to apoptotic cell death by DNA-damaging agents [53,58,59,60] and by blocking DNA double-strand break repair [42]. In view of the decreased expression of PARP1 and NFkappaB in several cancer cell types eradicated by PJ34, it has been suggested that eradication by PJ34 can be attributed to the attenuation of PARP1-dependent NF-kappaB activity that promotes proliferation [45]. However, the suggested mechanisms of PARP1-dependent activity of PJ34 in cancer cells are inconsistent with its exclusive PARP1 independent cytotoxic activity in human cancer cells at a higher concentration range than that causing PARP1 inhibition [29,30,31,35]. Its exclusive cytotoxic activity in human cancer cells was not shared by other potent PARP inhibitors [29,30,31,61]. PJ34 applied at the concentration range eradicating cancer cells did not cause breaks in the DNA, nor impaired healthy cells, and its activity was not restricted to cancer cells exposed to DNA-damaging agents nor BRCA mutants [28,29,30,31,32,35,46].

Flow cytometry measurements revealed that PJ34 exclusively arrests mitosis in human cancer cells [28,29,32]. Incubation with PJ34 causes cell-cycle arrest in a variety of healthy proliferating and cancer cells within 3–6 h [28]. However, healthy cells of mouse and human origin overcame the imposed cell-cycle arrest [28,29,31,33], and continued to proliferate in the presence of PJ34 similarly to untreated cells for weeks [28,29] (Figure 2). The tested healthy proliferating human cells included human epithelial, mesenchymal and endothelial cells [28,29,31]. Proliferating healthy cells of mouse origin included mouse embryonic fibroblasts, fibroblasts [28,29,32], and neuronal progenitor cells and astroglia [33].

In contrast, the cell-cycle arrest in mitosis which was imposed by PJ34 in human cancer cells was irreversible in all the tested human cancer cells, and the mitosis arrest was accompanied by cell death [28,29,30,31,32] (Figure 2). Cell-cycle arrest preceding cell death was measured in a variety of human cancer cell types, including cancer cells prepared from human colon, ovary, lung, pancreas, T-cell leukemia and breast cancers, including triple negative breast cancer [28,29,31,32,44].

Flow cytometry measurements at the measured time intervals did not reveal G1 or S-phase arrest in the tested human healthy and cancer cells treated with PJ34 (10, 20 and 30 μM) ([28,29], Figure 2)).

Since many of the tested malignant cell types included a high percentage of extra-centrosomal cells [28,29,30], the cytotoxic activity of PJ34 was attributed to de-clustering of extra-centrosomes in the mitotic spindle poles of multi-centrosomal cancer cells [29,30]. According to several reports, de-clustering of centrosomes in multi-centrosomal cancer cells leads to aberrant multi-polar spindles with un-assembled chromosomes [30,62,63,64]. Thus, it has been suggested that multi-centrosomes de-clustering in the spindle poles activates the spindle assembly check-point (SAC) proteins, which arrest mitosis [65,66], subsequently leading to mitotic catastrophe cell death [65,66,67,68].

Since mitotic arrest was induced by PJ34 in a variety of cancer cells, including cell types with a low percentage of extra-centrosomal cells [69,70], other mechanisms inducing mitotic catastrophe cell death were examined.

In one approach, all proteins currently known to be implicated in mitosis were screened in a group of human cancer and healthy cells, in an attempt to identify different effects of PJ34 (eradicating only cancer cells) on the post-translational modifications of proteins in the cancer versus healthy cells. Changes induced by PJ34 in their post-translational modifications were measured by the shift in their isoelectric point (IP) in two-dimensional (2-D) gel electrophoresis [31]. This approach was used because PARP inhibitors may modify a variety of proteins in both healthy and malignant cells [27,61,71].

Among other proteins associated with mitosis, proteins that are implicated in the structure of the mitotic spindle were examined for a possible interference with their post-translational modification by PJ34 in human cancer cells versus the effect of PJ34 on their post-translational modification in healthy epithelial cells [31]. This analysis was conducted in four types of human cancer cell lines (glioblastoma U87, PANC1 pancreas cancer cells, lung cancer cells A549 and triple-negative breast cancer cells MDA-MB-231). All these cells are eradicated by PJ34 [28,29,30,31,32]. A similar analysis was used to identify possible effects of PJ34 on the post-translational modifications of the same proteins in healthy human epithelial cells, which are not impaired by PJ34 [28,29,31].

This analysis identified only three proteins in the tested cancer cells with isoelectric point significantly shifted by PJ34, while not affected in healthy cells [31]. These proteins included two motor proteins [72], human kinesins 14/HSET/kifC1 and kif18A, and the non-motor protein NuMA (nuclear mitotic apparatus protein) [73,74,75,76,77]. For comparison, the IP shift of these proteins in the tested cells was also measured when these cells were incubated with the PARP inhibitor ABT-888 (Veliparib), which is a potent inhibitor of PARP-1,-2,-3 proteins (with a similar affinity for PARP1 and PARP2, *Kd* = 2.9 and 5.2 nM, respectively) [8,31]. Unlike PJ34, ABT-888 did not affect the isoelectric points of human kinesins 14/HSET/kifC1 and kif18A and of NuMA. There was no difference between the IP of these proteins in ABT-888-treated versus untreated cancer cells [31]. This result may exclude possible involvement of PARP-1,-2,-3 inhibition in the IP shift induced by PJ34 in the tested cancer cells [31]. In accordance, ABT-888 neither arrested mitosis, nor eradicated the tested human cancer cells [31]. These findings suggested a possible interference of PJ34 with the construction of the mitotic spindle in the tested human cancer cells.

Numerous findings indicate the essential role of HSET/kifC1 in the spindle structure of human cancer cells [73,74,75,78,79,80,81]. Differences in the expression and function of HSET in cancer versus healthy cells have been reported [74,78,79,80]. HSET/kifC1 inhibition or silencing causes small aberrant spindles in human malignant cells [79].

The kinesin Kif18A is implicated in microtubules de-polymerization, necessary for the binding of the duplicated chromosomes to kinetochores in the spindle mid-zone. Loss of its function results in the formation of long microtubules with chromosomes un-attached to kinetochores in the mid-zone [76].

The third identified protein, NuMA, is essential for mitosis in both malignant and benign cells [77,82,83]. Recently, several reports indicated some differences between NuMA proteins and their expression in benign versus malignant cells [84,85]. In this analysis, a clear-cut difference has been disclosed by the effect of PJ34 on the post-translational modification of NuMA only in the variety of human malignant cells ([31] and Figure 3). PJ34 did not affect the isoelectric point of NuMA in the healthy epithelial cells [31]. Concomitantly, the ability of NuMA to bind proteins was lost in the PJ34-treated malignant cells [31]. Moreover, the lost ability of NuMA to bind proteins was accompanied by un-clustering of NuMA in the spindle poles of the malignant cells treated with PJ34, as disclosed by confocal imaging [31] (Figure 3). In contrast, the bi-polar clustering of NuMA in the mitotic spindles of the healthy benign cells was not affected by the treatment with PJ34 ([31] (Figure 3).

Previous findings indicated the post-translational modification of NuMA by both polyADP-ribosylation and phosphorylation, both modifications promoting the binding of NuMA to proteins [82,83,84,85,86,87,88,89]. NuMA is phosphorylated by serine-threonine kinase pim1, and NuMA phosphorylation by serine threonine kinases at a specific site is crucial for its ability to bind proteins [87,89]. In addition, polyADP-ribosylation of NuMA by tankyrase1 in cancer cells promotes the ability of NuMA to bind other proteins [31,88].

Pim kinases and tankyrase1 are both inhibited by PJ34 at the concentrations range PJ34 causes cell death in human cancer cells (measured IC50 = 3.7 μM for pim1 inhibition by PJ34, and for tankyrase1 inhibition by PJ34, IC50 = 1 μM) [90,91,92]. Furthermore, tankyrase1 and pim kinases are hardly expressed in healthy somatic cells, while they are highly expressed in human cancer cells [92,93].

Clustered NuMA in the spindle poles and tethering of microtubules to the clustered proteins in the spindle poles are essential for the construction of stable poles required for the binding of the chromosomes to kinetochores in the spindle mid-zone [31,77,78,79,80,81,82,83]. Thus, PJ34 blocking of the post-translational modification of NuMA concomitantly prevents the clustering of NuMA in the spindle poles, causing mitotic arrest and cell death by preventing the binding of chromosomes to kinetochores in the spindle mid zone [31,65,66,67,68,80,81,82].

The causal association of the post-translational modification of NuMA with the ability of NuMA to bind proteins [31,87,88,89] could explain the interference of PJ34 with NuMA clustering and binding to HSET in the spindle poles of cancer cells treated with PJ34 [31,80,81,82,83]. NuMA clustering in the poles and the tethering of microtubules to the spindle poles implicates that HSET and NuMA binding in the protein clusters causes a stable structure that is lost in the spindle poles of human cancer cells treated with PJ34, at the concentration range inhibiting pim1 and tankyrase1 [31,91,92].

An unstable structure of the spindle poles due to impaired function of HSET and NuMA, and impaired function of kif18A in the binding of chromosomes to the kinetochores may result in scattered chromosomes instead of chromosomes bound to the kinetochores in the mid-zone (Figure 3). This activates the spindle assembly control (SAC) proteins, which leads to mitosis arrest followed by mitotic catastrophe cell death when the structural anomaly is not amended [65,66,67,68]. This is exactly the phenomenon observed by confocal imaging in a variety of human cancer cells treated with PJ34 [29,30,31]. De-clustering of centrosomes observed in multi-centrosomal cancer cells treated with PJ34 (20 μM) could result from the un-stable aberrant spindle poles [31,78]. Thus, prevention of the post-translational modification of NuMA and kinesins HSET and Kif18A by PJ34, could evoke mitotic arrest and mitotic catastrophe cell death in human cancer cells [30,31,32] (Figure 2) just by inserting specific anomalies in their mitotic spindle structure ([31] and Figure 3). This effect of PJ34 was not copied by ABT-888 at concentrations inhibiting PARP-1,-2,-3 [8,31]. It was achieved by inhibiting the activity of serine-threonine kinases, and by inhibition of tankyrase polyADP-ribosylation [31]. Other tankyrase inhibitors and tankyrase1 silencing caused similar faults in the structure of spindle poles as well as G2/M arrest, which could be attributed to inhibition of NuMA polyADP-ribosylation [31,88,94,95]. These results are consistent with the consequences of the treatment with PJ34 causing aberrant spindles with dispersed chromosomes (Figure 3) arresting mitosis and killing cancer cells without impairing healthy proliferating cells (Figure 2) [28,29,30,31,32]. In consistence, the findings of Leber and collaborators implicated NuMA and HSET in the construction of aberrant spindle poles with un-clustered multi-centrosomes in human cancer cells [96].

## 5. The Potency of PJ34 and Other PARP Inhibitors in Preventing Metastases

PJ34 and additional PARP inhibitors inhibit the activity of matrix metalloproteinase-2 in a range of concentrations higher than that inducing PARP inhibition [97]. The measured IC50 was about 56 μM for PJ34 inhibition of MMP-2 [97], lower than the IC50 values of other tested PARP1 inhibitors [97].

In healthy tissues, matrix metalloproteinases (MMPs) are key enzymes in the development and remodeling of tissues, including in wound healing [98]. However, MMPs are also dominant in tumor angiogenesis, in tumor cells escape from the primary tumor and their dissemination to secondary sites [98,99]. Elevated expression of MMPs, including MMP-2, has been implicated in metastasis [98,99,100]. Metastases predict high invasive stage of the malignancy, and a poor prognosis [99,100]. MMP inhibitors prevent metastases in a variety of solid cancer types [99,100]. MMP inhibition by PARP inhibitors is their additional advantage in cancer therapy. ABT-888 hindered cancer cell migration rate measured in vitro by the scratch assay [101]. PJ34 at the concentration range inhibiting MMP-2 prevented metastasis in melanoma xenografts [102].

## 6. Conclusions and Future Perspectives

The potency of PJ34 to exclusively eradicate human cancer cells without impairing healthy cells can be attributed to anomalies exclusively inserted in the structure of the mitotic spindle of human cancer cells. Blocking the post-translational modification of NuMA and kinesins HSET and kif18A by PJ34 causes these anomalies, which subsequently prevent the alignment of chromosomes bound to kinetochores in the spindle mid-zone. This structural anomaly in the mitotic spindle arrests mitosis and kills the cancer cell in the pre-anaphase stage by mitotic catastrophe cell death.

Notably, the modified phenanthridine PJ34, which has been invented for PARP inhibition, exclusively eradicates a variety of human cancer cells without impairing normal healthy somatic quiescent and proliferating cells. Thus, in spite of the permeability of PJ34 in the cell membrane and its rapid distribution in the animal’s tissues, treatment with PJ34 did not impair healthy tissues in the tested animals, nor their development and weight-gain.

On the basis of these findings, we hope that cell death evoked by structural faults in the mitotic spindle of human cancer cells will pave the way to a new concept in cancer therapy. Inserting specific structural anomalies in the mitotic spindle of human cancer cells may specifically eradicate cancer cells while saving healthy cells and physiological functions frequently lost during the currently offered DNA-damaging cancer therapies.

## Figures and Tables

**Figure 1 cancers-12-01628-f001:**
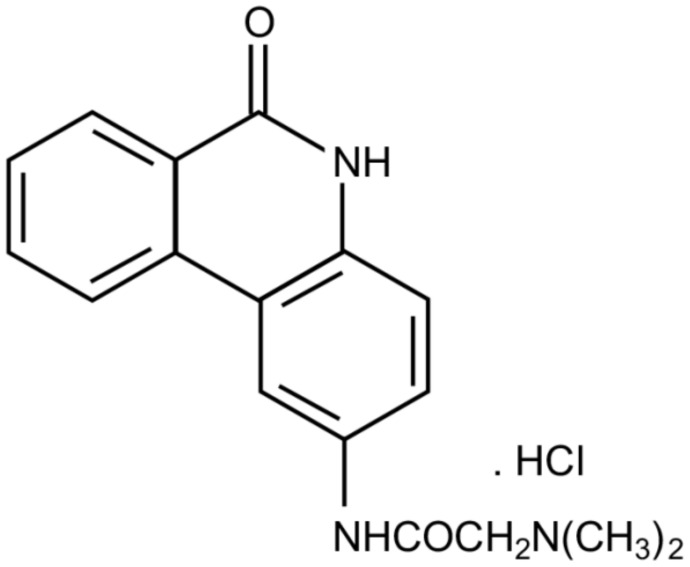
The chemical structure of PJ34, *N*-(6-Oxo-5,6-dihydrophenanthridin-2-yl)-(*N*,*N*-dimethylamino)acetamide hydrochloride.

**Figure 2 cancers-12-01628-f002:**
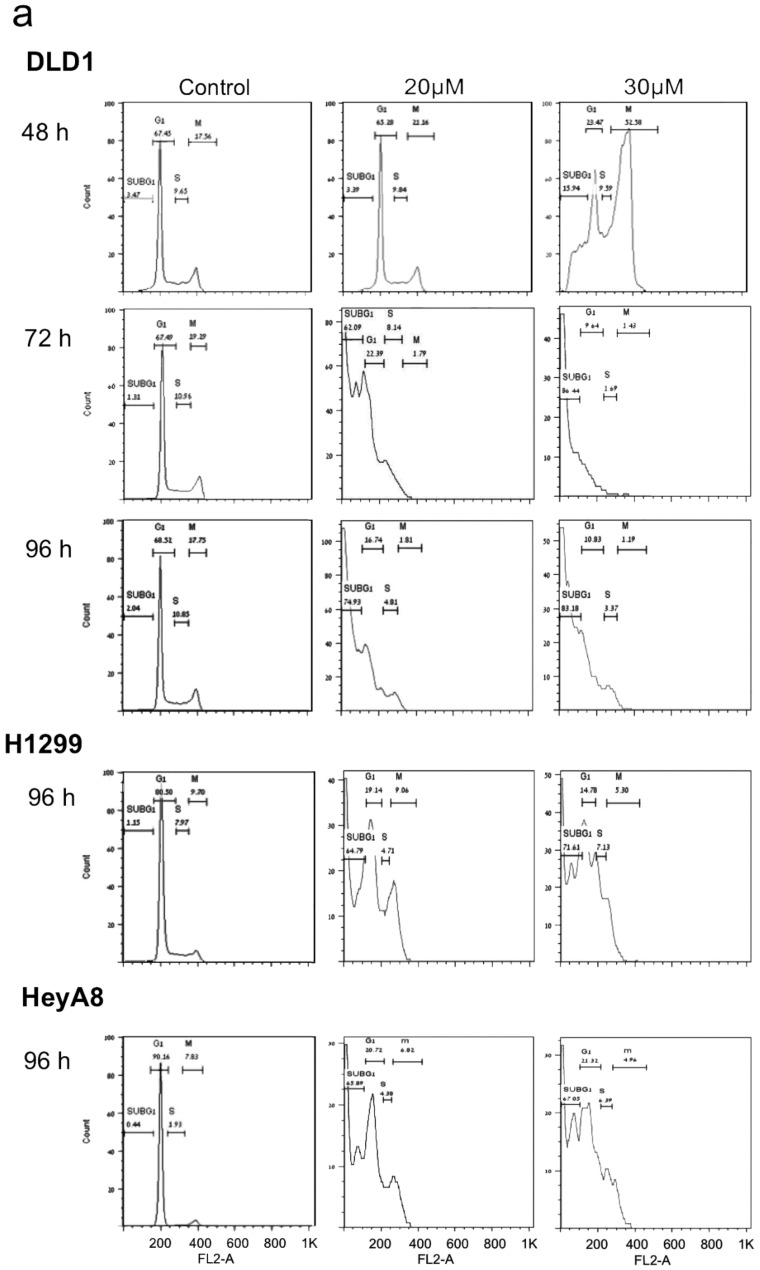
G2/M arrest and cell death in human cancer cells treated with PJ34. PJ34 does not impair the cell-cycle of human healthy proliferating cells. Flow cytometry of the indicated human cancer cells (**a**) and human healthy proliferating cells (**b**) is displayed. Cells were treated with PJ34 at the indicated concentrations and incubation periods. From: Castiel et al., 2011, *BMC Cancer*.

**Figure 3 cancers-12-01628-f003:**
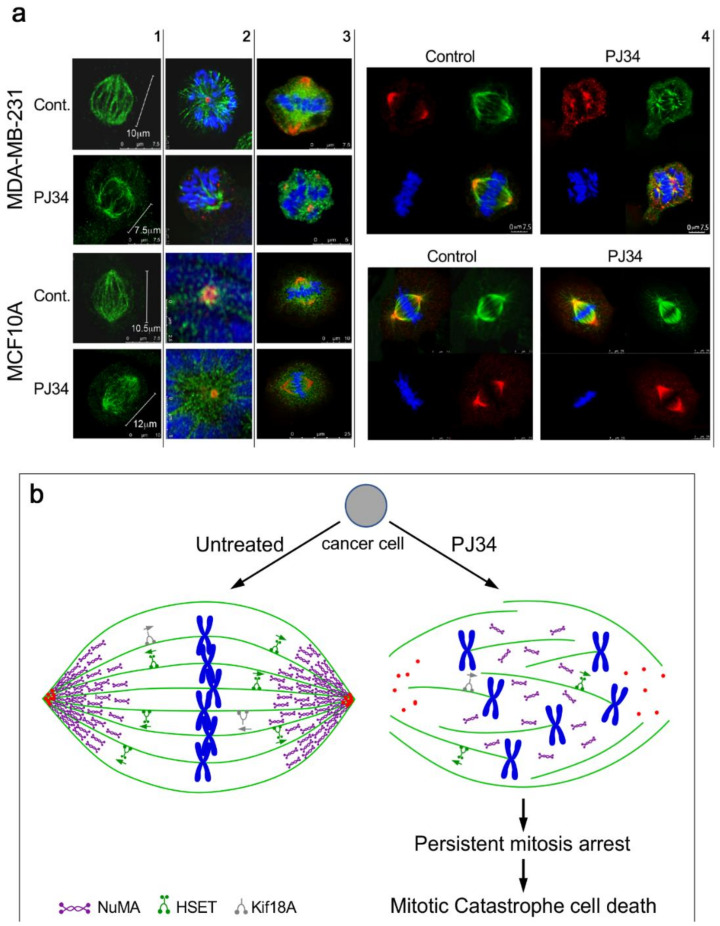
(**a**) Confocal images of mitotic spindles in human triple negative breast cancer cells (MDA-MB-231) and in human healthy breast epithelial cells (MCF10A), untreated or incubated with PJ34. Incubation of human cancer cells with PJ34 (20 μM, 27 h) impaired spindle poles (labeled by immunolabelling, γ-tubulin in the centrosomes—red), microtubules (labeled by immunolabelling—kinesin HSET or by immunolabelling α-tubulin—green), segregation and alignment of chromosomes (labeled by DAPI—blue), and NuMA clustering in the spindle poles (Immunolabeled NuMA—red). **Column 1**: Microtubules in spindles of healthy and cancer cells immunolabeled by the kinesin HSET in cancer and healthy cells, untreated and treated with PJ34. **Column 2**: Spindle poles labeled by *γ*-tubulin in healthy and cancer cells, untreated and treated with PJ34. HSET is immunolabeled in the microtubules. **Column 3**: Clustered NuMA in bipolar spindles of healthy cells either treated or not with PJ34, and in untreated cancer cells. Un-clustered NuMA in spindles of cancer cells treated with PJ34. **Column 4**: upper frame: In cancer cells—clustered NuMA in spindle poles and aligned chromosomes in the midzone of untreated cancer cells. Aberrant spindles, un-clustered NuMA and scattered chromosomes in cancer cells treated with PJ34. Lower frame: In healthy cells—clustered NuMA in the spindle poles and segregated chromosomes aligned in the mid-zone of the mitotic spindle of healthy cells either untreated or treated with PJ34. (**b**) A schematic presentation indicating the effect of PJ34 on the spindle structure in human cancer cell. In the untreated cancer cell, normal bipolar spindles with clustered NuMA, clustered multi-centrosomes, and aligned chromosomes in the spindle mid-zone. In the PJ-34 treated cancer cell, aberrant microtubules (**green**), aberrant spindle poles, un-clustered NuMA (as indicated), dispersed chromosomes (**blue**) and un-clustered multi-centrosome (**red**), From: Visochek et al., 2017, Oncotarget.

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
