# Peer review of "The Modified Phenanthridine PJ34 Unveils an Exclusive Cell-Death Mechanism in Human Cancer Cells"

_cancers, 2020, doi:10.3390/cancers12061628_

Round 1

Reviewer 1 Report

I do not have any further comments.

Reviewer 2 Report

The author satisfied my points satisfactorily.

Reviewer 3 Report

The author has adapted the review in a very nice manner. It now nicely summarises that although originally used as PARP inhibitor, PJ34 has other mechanisms of action that may be highly relevant. The review is very informative to researchers interested in the mechanism of PJ34.

The only comment I still have concerns figure 2: I don't see the added value of displaying these FACS results in this manner in a review.

This manuscript is a resubmission of an earlier submission. The following is a list of the peer review reports and author responses from that submission.

Round 1

Reviewer 1 Report

The author has focused on PJ34 as molecule destabilizing the mitotic spindle, leading to death of cancer cells specifically. The author first described a selection of PARP1 functions and regulatory mechanisms, however, further on it becomes clear that the observed cell death is most likely not due to PARP1 inhibition, as more specific PARP1 inhibitors do not show the same effect. The author then continues to review the effect of PJ34 on cultured cells as well as in mouse models and finishes with potential molecular mechanisms.

I am not sure whether this is a very useful review when regarding PJ34 as PARP1 inhibitor, as many generations of better, more specific inhibitors have followed PJ34, such as first olaparib and later for example veliparib. PJ34 is not a good inhibitor as it is not selective enough (see for example Wahlberg 2012 for original data or Slade 2020 for a review of PARP family inhibitors).

However, if the focus is shifted towards PJ34 as cell-death inducing agent, through further to be defined pathways, it might be useful. PJ34 has shown exciting effects in the studies performed by the author, but due to the focus on PARPs and sometimes apparently contradicting statements, it is difficult to follow the flow of the review. I would recommend to consider the following structure:

  • Intro historically PJ34 as PARP1 inhibitor: not very specific, discontinued in favor for more specific PARP1 inhibitors
  • Describe the effects of PJ34 in mouse models and tissue culture, while making it clear in which cell types and models it is showing effects
  • Discuss the potential molecular targets: PARP1, pim kinase, tankyrase, MMP1, others?
  • Outlook for future studies: finding the molecular targets of PJ34, determining toxicity for healthy cells, potential combinations with other therapeutics, ...

Specific comments:

  • Line 20: The most abundant nuclear polyADP-ribose polymerase, PARP1, catalyzes a post translational modification of proteins (polyADP-ribosylation), assembling ADP-ribose polymers on certain residues (serine, arginine, cysteine, lysine, and aspargine) of nuclear proteins [3].

This statement is not correct. When bound to co-factor HPF1, PARP1 can modify serines. When alone, it appears to modify acidic residues. Lysines have been shown to be an artifact; arginines are modified by ARTCs. As far as I am aware, to date only toxins have been demonstrated to modify cysteine or aspargine, not eukaryotic transferases.

The reference cited here is from 1994; there are many new discoveries since then, including the latest Nature article by the Ahel lab on a PARP1/HPF1 structure.

In general, many more recent works on PARP1 and poly-ADP-ribosylation were omitted that the author may want to reconsider including.

  • Ref 16 appears misplaced, as it is a review article about CTCF, without apparent relevance for PARP1-mediated chromatin remodeling?

  • Line 35: The author alludes to other PARP1 activation mechanisms besides damaged DNA. It would be helpful to the non-expert reader to briefly summarise these, where possible including the original research identifying these signals, as was done for activation of PARP1 in response to Ca-release and by binding to phosphorylated Erk. Alternatively, the author might explain why specifically calcium release and regulation through Erk binding were explained. Are they relevant for the inhibitor, but other activation mechanisms not? It might be beneficial to include an image summarizing these PARP1 regulatory signals and their consequences.

  • Line 77: “exclusively acting in human cancer cells”. It would be good to clarify better which cells are affected: are non-transformed human cells sensitive, transformed cells, which mouse cells? The authors do state that healthy human cells undergo a transient cell cycle arrest. Is the same true for mouse cells? If mouse cells do not respond at all, it makes the mouse work summarized in paragraph 4 questionable: if for example its target is not expressed in mouse cells, then potential cytotoxic side effects are not measured in this model. Kurowaka and colleagues (2019) for example observe that PJ34 suppresses the cell cycle in neural progenitor cells but not MEFs. This reference is missing altogether.

  • In Paragraph 2: how do non-cancer cells respond to these relatively high doses of PJ34 inhibitor? It would be good to include the information whether effects were measured on these controls. The fact that other PARP inhibitors do not show the same effect and the effect is independent of PARP1 expression and activity, implies that the inhibitor targets something other than PARP1 for its cell-death inducing effect. This needs to be clear throughout: is the author regarding PJ34 as PARP1 inhibitor causing cell death, or as potentially targeting other proteins to mediate these effects? As indicated above, the answer to this question might shift the focus of this review.

  • Paragraph 3 might be summarized in a table, displaying at one glance in which animal models with cancer cells were tested, which concentration inhibitor and what the outcome/side effects were.

  • Figure 2 is not readable. The author should consider to summarise the data differently to enhance effectiveness.

  • An error was made with refs 54/55, leading to a shift in their numbering.

  • Line 244: “Pim kinases and tankyrases are both inhibited by PJ34”. For the tankyrases, highly specific inhibitors are available such as XAV939. Do they copy the phenotypes observed with PJ34? Taking the promiscuity of this inhibitor into account, I would recommend restructuring the review as recommended above. As it is highly likely that the measured effects on cell-growth are independent of PARP1, I recommend to trim the parts of the review describing PARP1, especially as many recent discoveries are missing in this text, but instead refer to another review such as the excellent recent review by Slade Genes & Dev 2020. 

  • The author may consider to graphically display the different targets of PJ34 and how they could contribute to the induced cancer cell death.

  • Reading the review left me wondering whether PJ34 would be suitable as therapy, as it has so many potential targets with undisclosed downstream effects. It would be good to include a little discussion on this topic, and include information on potential clinical trials – have any been attempted?

Reviewer 2 Report

The review article by Malka Cohen-Armon summarises well the antitumor properties of the small molecule PJ34.

Major points:

  • PARP1 is just one of the PJ34 targets. Contrary to the new generation of highly selective inhibitors, the PJ34 (which belongs to the old class of PARP inhibitors) binds and inhibits the almost entire range of catalytically active PARPs, which means Tankyrases and even mono(ADP-ribosyl)transferases. Not surprisingly, the major cytotoxic effects of PJ34 are obtained at a high concentration of inhibitor; for instance, the cell cycle outcomes of PJ34 are compatible with the inhibition of Tankyrases. In my opinion, the background of the manuscript seems too centered on PARP1, in contrast, very little is about other PARP substrates. Thus, it would be helpful for readers to obtain a balanced intro that summarises the known cellular functions of all PARPs targeted by PJ34.
  • Furthermore, a few words about new classes of specific inhibitors (e.g. new Tankyrases, PARP3, PARP11, and PARP10 inhibitors) would be really appreciated by the community.
  • A table summarising the reported IC50 of PJ34 for diverse PARPs and not-PARPs would be extremely useful in order to get the feeling of which cellular phenotypes may, or not may, be dependent on PARPs (and, if so, by which of them) or by kinase inhibition.
  • Regarding the mechanism of action of PJ34 in human cancer cells, it should be considered the effect of PJ34 on other PARPs (g.tankyrases and PARP3, both have described mitotic functions). Could the author comment on the possibility that the enrichment of PJ34-treated cells in G2/M phases of Cell Cycle may be also due to genomic abnormalities accumulated during uncontrolled S-phase in cancer cells (perhaps p21/p53/ATM/ATR/BRCA-deficient), thus preventing the correct mitotic entry and completion compared to proficient cells?
  • Regarding NUMA1, the author may wish to further comment the relevant literature on NuMA1 mitotic regulation by PARPs (Tankyrase and PARP3). Also, the author should mention in few words that additional kinases (AURKA, CDK1, ATM) have been described as phosphorylating NuMA1; the inhibition of those kinases leads to overlapping phenotypes. Does PJ34 have affinity for any of these reported kinases?

Minor points:

  • Page1, line 18: “…its activity as a PARP1 inhibitor”, please change to “PARP inhibitor”.
  • Page 1, lines 21-22: please cite more relevant and recent articles regarding amino acids modified by PARPs.
  • Page 2, line 42: ”…This signal-induced PARP”, please specify which PARP the author is referring to.
  • Page 2, line 53: Please change “NAD” to “NAD+”.
  • Page 5, line 184: please change “includ” to “includes”.

Reviewer 3 Report

The submitted paper describes basically only one mechanism of PJ34-induced inhibition of the cancer cell growth.  The major finding is preceded by extensive introduction that aims to disclose the premises and observations from cell cultures and xenografts, which lead to discovery of the PJ43 effects on the mitotic spindle. Relatively long text can be summarized in 2-3 sentences.  Therefore, I suggest to include more information on:

  1. The current state of clinical or preclinical trials on PJ34. Since such a study has not been registered yet (according to clinicaltrals.org) then what is an obstacle of using this compound in anti-cancer therapies? Especially, when other PARP inhibitors such as olaparib has been approved by FDA for the treatment of some cancer types.
  2. Although PJ34 only transiently inhibited proliferation of mesenchymal stem cells, it has been reported to interfere with the formation of osteoblasts. Some potential side effects of PJ34 should be mentioned.

I do not mind including the published figures or pictures as long as such an approach is accepted by the journal. However, instead of copying and pasting the Author could present the relevant findings (alterations in cell cycle progression) in a scheme or do over available histograms to e.g. column chart representing quantitative cell distribution in particular phases and how these values are affected by PJ34. In the current form the included graphs are unreadable.

As for graphs, the small drawing placed below confocal pictures does not explain in detail the action of PJ34. More advanced and molecular approach (e.g. mitotic chromosomes with the focus on spindle pole, local – interaction with NuMA and global – chromosome segregation, growth arrest, death, effects of PJ34 ) is needed to make the described mechanism intelligible also for wired audience.

Minor points – more attention should be paid to writing; e.g.

  1. “Eradication” appears too many times in the text. It can be replaced by decreased number, toxicity, inhibited growth etc
  2. Again, example of repetitive wording: “Unexpectedly, this research disclosed a promising cell-death mechanism, exclusively acting in human cancer cells [40]. This mechanism has been disclosed via the cytotoxic activity of a group of phenanthrene derivatives acting as PARP inhibitors [40-42].

Since the manuscript describes only one mechanism I would recommend transferring it to perspectives and shortening to the key aspect - PJ34 action in spindle pole of cancer cells and future perspectives and limitations.